

# Pilot study of locomotor asymmetry in horses walking in circles with and without a rider

Agneta Egenvall[1], Hilary M. Clayton[2] and Anna Byström[3]

[1] Swedish University of Agricultural Sciences, Department of Clinical Sciences, Uppsala, Sweden
[2] College of Veterinary Medicine, Michigan State University, Department of Large Animal Clinical Sciences, East Lansing, MI, United States of America
[3] Swedish University of Agricultural Sciences, Department of Animal Environment and Health, Uppsala, Sweden

Corresponding author
Agneta Egenvall,
agneta.egenvall@slu.se

## ABSTRACT

**Background:** Horses commonly show asymmetries that manifest as left (L)-right (R) differences in vertical excursion of axial body segments. Moving on a circle confounds inherent individual asymmetries. Our goals were to evaluate individual and group asymmetry patterns and compare objective data with subjective impressions of side preference/laterality in horses walking on L and R circles.
**Methods:** Fifteen horses walked on L and R circles unridden and ridden on long and short reins. Optical motion capture (150 Hz) tracked skin-fixed markers. Variables were trunk horizontal angle; neck-to-trunk angle; vertical range of motion (ROM) for the head, withers and sacrum; ROM for pelvic roll, pitch, and yaw; mean pelvic pitch; and ROM for hip, stifle and tarsal joints. Differences between inside and outside hind steps were determined for vertical minima and maxima of the head (HMinDiff/HMaxDiff), withers (WMinDiff/WMaxDiff) and sacrum (PMinDiff/PMaxDiff). Subjective laterality was provided by owners. Data analysis used mixed models, first without and then with subjective laterality. Iterative k-means cluster analysis was used to associate biomechanical variables with subjective laterality.
**Results:** PMaxDiff, PMinDiff and WMaxDiff indicated R limb asymmetry in both directions. WMinDiff indicated L (inside) fore asymmetry for L direction but was close to zero for R direction. Hip ROM was significantly smaller for the inside limb in both directions (L inside/outside: 16.7° *vs.* 20.6°; R: 17.8° *vs.* 19.4°). Stifle ROM was significantly larger for the inside limb in both directions (L: 43.1° *vs.* 39.0°; R: 41.9° *vs.* 40.4°). Taking the general direction effect into account the R hip and L stifle had larger ROM. Adding laterality to the models (seven horses L- *vs.* six horses R-hollow), PMaxDiff R hind asymmetry was more obvious for L-hollow horses than for R-hollow horses. L-hollow horses had greater pelvic roll ROM moving in L *vs.* R direction. L-hollow horses had smaller inside and greater outside hip joint ROM in L *vs.* R direction. R-hollow horses had a significant difference in HMinDiff between L (0 mm) and R (−14 mm) directions, indicating less head lowering at outside forelimb midstance in R direction, and larger outside tarsal ROM in R (38.6°) *vs.* L (37.4°) direction ($p \leq 0.05$). The variables that agreed most frequently with subjective laterality in cluster analysis were pelvic roll ROM, followed by HMinDiff and PMaxDiff.

**Conclusion:** Differences between horses walking in L and R directions were found both at group and individual levels, as well as evidence of associations with subjective laterality. Horses maintained more symmetric hip and stifle ROM and withers vertical motion when walking on the R circle. Findings suggest that left and right lateralised horses may not be perfect mirror images. Pelvic roll ROM emerged as a promising variable to determine laterality in walk as perceived by the rider, especially when considered together with other variables.

## INTRODUCTION

Laterality describes dominance of one side of the brain in controlling the function of paired body parts, resulting in a functional side preference. Laterality can be present at individual or population level (*Rogers, 1989*). Laterality at individual level implies that individuals have a left or right asymmetry pattern or preference but does not imply a consistent bias in the population as a whole. Population-level laterality exists when a majority of the population is biased towards the same side. In people, 90% are right-handed and 10% left-handed illustrating a marked population-level bias (*Papadatou-Pastou et al., 2020*). For the human species leg dominance has been associated with milder population bias: in one study 62% were found to be right-legged, 8% left-legged and 30% mixed-legged (*Tran & Voracek, 2016*). In horses, there is evidence of sensory (*e.g., McGreevy & Rogers, 2005*; *Farmer et al., 2018*) and motor (*e.g., Colborne, Heaps & Franklin, 2009*; *Lucidi et al., 2013*; *Byström et al., 2018*) asymmetries that may be due to laterality.

Motor laterality has been documented in many species (*Rogers, 1989*). In horses, asymmetries thought to be associated with motor laterality have been reported in foals and unhandled youngsters (*Drevemo et al., 1987*; *Van Heel et al., 2006*; *Lucidi et al., 2013*), and it has been suggested that the degree of asymmetry increases with age (*McGreevy & Thomsen, 2006*; *Lucidi et al., 2013*). It is also generally accepted among equestrians that horses are inherently crooked and one of the tasks addressed during training is to straighten the horse, *i.e.*, teach the horse to use the left and right sides of the body more symmetrically (c.f. *Byström et al., 2020*). In equestrian terminology, a horse is described as being "straight" when the hind limbs follow the tracks of the forelimbs. On curved lines this implies a degree of spinal lateral bending. When the hind limbs do not follow the tracks of the forelimbs, the horse is described as being "crooked".

While scientists and equestrians agree that motor laterality is likely to be present in horses, at least to some extent, the pattern of asymmetries described overlap only partially between equestrians' perceptions and the scientific literature. Equestrians frequently describe a difference in the horse's ability to turn in left *vs.* right direction (*Murphy & Arkins, 2008*; *Kuhnke et al., 2010*; *Kuhnke & König von Borstel, 2022*). One side is described as the "hollow" side based on the horse bending more easily towards that side and the other side described as the "stiff" side due to the horse's reluctance to bend towards that side (*Byström et al., 2020*). The rider usually perceives that the horse accepts greater rein

contact on the stiff side, but this may be confounded by rider handedness (*Kuhnke et al., 2010*; *Hawson et al., 2014*; *Kuhnke & König von Borstel, 2022*). When circling, the horse drifts towards the stiff side in both directions, such that the hind limbs do not follow the tracks of the forelimbs. Other aspects of asymmetry may be evident by comparing spatiotemporal kinematics of contralateral limbs; a rider may for example describe that one hind limb takes shorter steps. Scientific studies have described that many foals have a preferred limb position when grazing with one forelimb protracted and the other retracted (*Van Heel et al., 2006*) and this finding has been applied in the development of behavioural tests for limb preference (*Kuhnke et al., 2010*; *Shivley, Grandin & Deesing, 2016*). However, mature feral horses do not show a side preference for forelimb protraction during grazing (*Austin & Rogers, 2007*). The role of asymmetry in a horse's fear and flight responses has also been studied (*Larose et al., 2006*; *Austin & Rogers, 2007*; *Sankey et al., 2011*; *Siniscalchi et al., 2014*).

At present, it is unclear to what extent these laterality patterns are associated with the asymmetries commonly described by equestrians. Further, asymmetries or side preferences may, apart from laterality, also be related to other factors, such as past or present injuries, habit, and human influence (*Byström et al., 2020*). In general, research findings suggest the presence of laterality in horses (*Byström et al., 2020*), however, the majority of studies addressing (a)symmetry of locomotor performance have been directed towards pathological rather than functional causes. In lame horses, kinematic asymmetries have a pathological basis associated with pain, neurological dysfunction, or movement restriction and the locomotor asymmetries are adopted to reduce loading of the lame limb(s). These asymmetries are usually evaluated during trotting and are measured in terms of asymmetrical vertical displacements of the head, withers and pelvis on the left and right diagonals (*Davidson, 2018*; *Reed et al., 2020*). However, a recent study describes a weak association between rider-perceived sidedness to push-off lameness in trot (*Leclercq et al., 2023*). Much less is known about movement adaptations in lame horses at the walk. Vertical movement asymmetry of the head and withers have been described in horses with induced forelimb lameness walking on a treadmill (*Buchner et al., 1996*; *Serra Bragança et al., 2021*). There is a need for scientific evidence to clarify the relationships between the horse's inherent asymmetry patterns, in the context of the equestrian experience, to understand and measure the horse's inherent crookedness scientifically.

To study motor laterality objectively, it may be necessary to evaluate several, multi-facetted variables in space and time with sufficient accuracy to detect subtle left-right differences. For example, in a study of kinematic asymmetries in walk it was shown that one hind limb may be less protracted and the hoof was placed more medially relative to the trunk than the contralateral hind limb (*Byström et al., 2018*). The temporal relationship between the limbs may differ with the movements occurring slightly earlier on one side compared with the other. Few methods of analysis offer sufficient precision within a large study volume to measure and define such variables. For measuring spatial relationships, for example between the limbs, the best option is optical motion capture as inertial measurement units cannot, as yet, measure distances between sensors with sufficient accuracy. The other challenge is determining whether the measured asymmetries do

indeed reflect motor laterality. It is well known that a large proportion of supposedly sound riding horses display asymmetries of a magnitude that clearly overlaps with low-grade lameness (*Rhodin et al., 2017*; *Hardeman et al., 2022*). Completely excluding lameness as the cause of an observed asymmetry in a study group of horses is difficult. One way this problem has previously been addressed is by confirming that vertical movement asymmetries are not increased from walk to trot (*Byström et al., 2021*), which would be expected in a lame horse.

In this pilot study we target the issue of inherent asymmetries that are addressed by equestrians on a daily basis as they strive to make their horse straighter. Because relevant biomechanical evidence is scant, we chose to perform a methodological study on a smaller population of horses in order to inspire work in this area. The aim was to study asymmetry in horses walking around circles to the left and right using optical motion capture and contrast the findings to owner-perceived laterality while training. As it is often debated whether the presence of a rider is associated with the horse becoming less or more crooked (*Byström et al., 2021*), horses were measured both with and without a rider. Variables targeted were vertical excursions of the head, withers and pelvis, pelvic rotations (roll, pitch and yaw), and hind limb joint angles, neck-trunk angle, and orientation of the trunk relative to the direction of travel (trunk horizontal angle). While the primary goal was to describe patterns that were common across horses, individual patterns were also assessed during this attempt to unravel kinematic patterns of motor laterality in the horse.

## MATERIALS AND METHODS

### Horses

The study included 15 horses of various breeds and sizes (five mares, eight geldings, two stallions; median age 11 years with range 6–24 years) housed at the same stable (Table S1). All horses were unshod and were being actively trained at various levels of classical dressage. None was used for competition. All owners considered their horses to be sound. The horses were evaluated for soundness by a veterinarian (AE) in-hand and on the lunge on a soft surface and all were deemed sound in trot and showed normal back function.

According to Swedish law, ethical approval is not required for non-invasive experiments that don't put the animals at any greater risk than during their normal daily activities. Horse owners gave written informed consent for the data collection.

### Riders/handlers and subjective evaluation of the horses' laterality

Each horse was handled and ridden by one of seven participating riders, who were familiar with the horses. There was one male (height: 1.90 m; weight: 85 kg) and six female (height: 160–173 m; weight: 54–67 kg) riders aged 18–52 years. All riders considered themselves right-handed.

A questionnaire (Table S2) was formulated for subjective assessment of the horse's crookedness, that is, which side the rider considered to be the horse's stiff side and hollow side, or if the horse was perceived as symmetric. It included the following concepts:

- which direction (if any) does the horse tend to fall to the outside when turning or circling,
- which direction (if any) does the horse tend to fall into the circle,
- which direction the horse was easier to bend to—it was carefully explained to respondents that this meant which side was easier, even if the bend was not optimal (*e.g.*, tendency to over-bend),
- on which rein does the horse accept greater rein contact (regardless of direction of movement).

These questions were asked verbally, and free text answers were recorded when relevant. Hollow side was defined as the side where the rider felt lower rein tension and found that bending was easier and that the horse drifted out of the circle by falling out over the outside shoulder. If the horse followed this pattern for either direction, that direction was assigned the horse's hollow side. For each horse, the questionnaire responses did not always follow the expected pattern for all questions, and hollow side was then determined by weighing the answers together. Agreement *vs.* disagreement between the questionnaire responses and the expected response according to the assigned hollow side for each horse can be found in Table S2.

## Markers

Spherical 25 mm reflective markers were attached to the horse with double-sided adhesive tape. Markers used in the present study were located at the poll (midline just behind the ears), top of the withers (T6), the lumbosacral joint, left and right tuber coxae, hip joint (anterior part of the greater trochanter of femur), stifle joint (just caudal to the distal attachment of the lateral collateral ligament of the femorotibial joint), the tarsal joint (laterally on the talus), and the lateral condyle of the third metatarsal bone (Fig. 1).

## Data collection

Data were collected in a 20 * 30 m indoor arena with footing composed of sand and synthetic fibres. High-speed infrared cameras (Oqus 700+a), sampling at 150 Hz, were arranged around the arena. The measuring volume was approximately 10 * 10 * 3 m, which was the maximal volume that could be covered by the available cameras. Ground poles were laid out in a square to indicate the extent of the volume. On each collection day, one or two horses were measured after dusk, when there was no sunlight to interfere with the motion capture. Ambient temperatures were −5 °C to +5 °C. Calibration of the data collection volume was repeated daily with the criterion for acceptance being an average calibration residual <3.0 mm, otherwise the calibration was repeated. Data collections were also recorded on video (Sony FDR-AX53; Sony, Minato City, Tokyo, Japan) at 25 Hz.

The horses first performed a set of unridden exercises. The horses were walked in hand in a straight line and in left and right circles, and then lunged to the left and to the right, wearing a cavesson with the lunge line attached mid-dorsally. After this the horses were saddled and bridled, either with a bit or a bitless bridle depending on what was regularly used for each horse (Table S1). After a short warm-up, horses were ridden in walk in

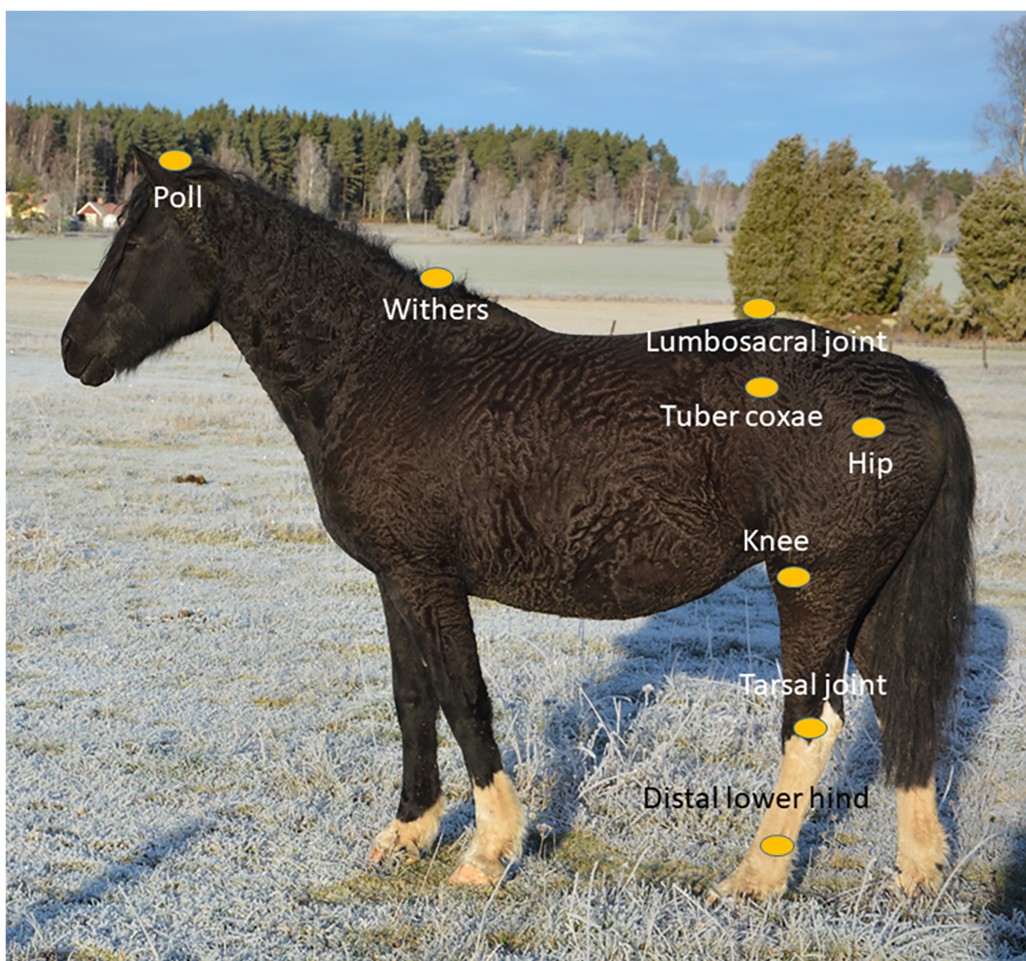

**Figure 1 Marker placement.** Markers were placed at the poll, the highest point of the withers (T6), at the lumbosacral joint (LS), left and right tubera coxae (TC), over the knee, stifle and tarsal joints and over the laterodistal part of the third metatarsal bones.

straight lines and on left and right circles, first on long reins and then on a contact (with shortened reins). Circle size was ~9 m throughout, limited by the size of the measuring volume. Data were collected for two complete circles for each direction and condition. All horses performed the exercises at a comfortable speed, taking care to maintain consistent speed between directions. The direction (left, right) of the first circle was alternated each day with eight horses starting to the left and seven horses starting to the right. Only data collected on the circle in walk were used in the current study. Data both from walk in hand and from lungeing were labeled as unridden condition.

For the limb joint ROM variables, strides were only included if the ROM value was within the following limits: for the hip, strides were included if the ROM was >10° and <33°, for the stifle if the ROM was >25° and <57° and for the tarsus if ROM was >25° and <55°. These limits were determined based on scatterplots of the data and previously published data for hind limb joint ROM in walk (*Hodson, Clayton & Lanovaz, 2001*). For the remaining variables, strides with head vertical range of motion outside ±40% of the

trial mean vertical head range of motion, with pelvic vertical range of motion outside ±20% of the trial mean vertical pelvis range of motion, and/or strides with a stride duration outside ±20% of the trial mean stride duration were automatically removed, in order to exclude strides where the horse was not in steady-state locomotion (*Hardeman et al., 2022*).

## Data analysis

Scripts were written in Matlab (version R2020a; Mathworks, Natick, MA, USA) for analysis of kinematic data, producing time-series variables (see below). Data were processed as previously described in *Egenvall, Engström & Byström (2023)*. In short, circle radius was determined for each measurement (trial) through fitting a circle to the x and y (horizontal plane) coordinate data from the lumbosacral joint marker using the least squares method. Strides were segmented at maximal protraction of the inside hind limb. Hind limb protraction-retraction angles were calculated as the angle between a line connecting the withers marker and the lumbosacral joint marker, and a line between the lumbosacral joint marker and the hind cannon marker. Protraction-retraction data were band-pass filtered using a zero-lag Butterworth filter with cutoffs at 0.5 and four times the stride frequency, to facilitate identification of extreme points. Hind limb maximum protraction was then identified. The kinematic variables were time-normalised to 0–100% (201 values per stride) before extraction of data for statistical analysis (*Egenvall, Engström & Byström, 2023*).

Speed was determined from the movement of the lumbosacral joint marker in the horizontal plane. The variable 'trunk horizontal angle' which describes the orientation of the horse's body in the horizontal plane, was calculated as the angle between the direction of movement (velocity vector) of the lumbosacral joint marker and a line connecting the withers and lumbosacral joint markers, with positive values assigned when the hindquarters were to the right of the forehand in the direction of motion. The variable 'neck-trunk angle', representing the neck angle and head position relative to the trunk, was calculated as the angle in the horizontal plane between a line connecting the poll and withers markers and a line connecting the withers and lumbosacral joint markers. Neck-trunk angle was positive when the head was to the right of the body axis in the direction of movement. Stride mean was determined for trunk horizontal and neck-trunk angles.

Pelvic roll (rotation around the long axis of the body) was measured relative to the horizontal, based on the markers on the left and right tuber coxae. Pelvic pitch (rotation around the transverse axis) was based on the lumbosacral junction marker and the average position between the markers on the two tubera coxae. Pitch was expressed relative to a line joining the withers and lumbosacral joint markers. Positive pitch was defined as clockwise rotation when viewed from the right, *i.e.*, raising the base of the tail relative to the lumbosacral junction (suggestive of lumbosacral extension). Pelvic yaw (rotation around the vertical axis) was calculated based on the tuber coxae markers, relative to a line between

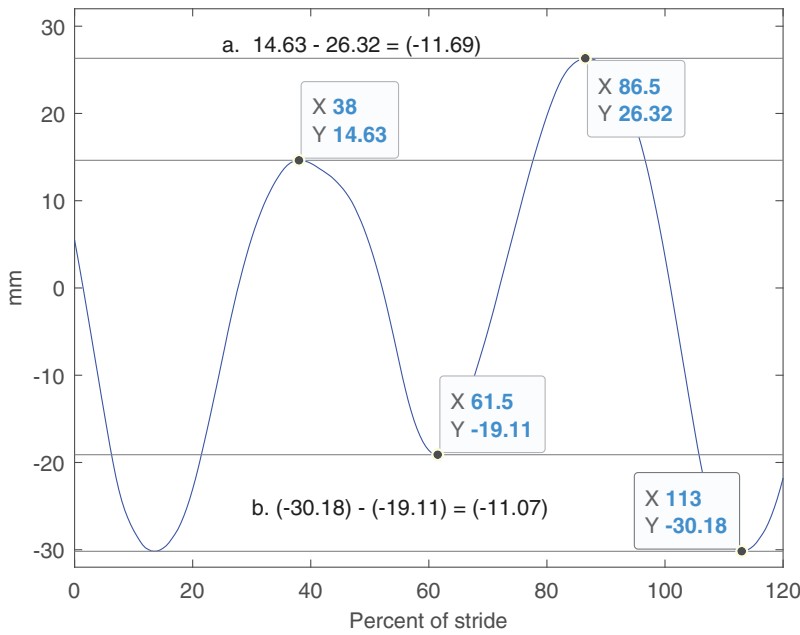

**Figure 2 Schematic example of calculations of minimum and maximum vertical differences.** Shown is pelvic vertical motion for a stride starting at left hind limb maximum protraction. In calculation a. (MaxDiff) the maximum at right hind midstance is subtracted from the left hind maximum. In the example this yields a negative MaxDiff, *i.e.*, the horse croup is lower at left hind midstance. In b. (MinDiff) the minimum during late left hind stance is subtracted from the corresponding right hind minimum. In the example this yields a negative MinDiff, *i.e.*, the croup is relatively higher at the end of left hind stance. Zero and 100% of the stride corresponds to maximum inside hind limb protraction and hind limb ground contact generally occurs 6–7% after maximum protraction (*Hodson, Clayton & Lanovaz, 2001*).

the withers and lumbosacral joint markers. From the pelvic rotation data, stride ROM for pelvic roll, pitch, and yaw and stride mean pelvic pitch were determined.

Vertical motion symmetry variables were measured as the difference between the left and right hind limb steps in the minimum and maximum heights of the head (HMinDiff HMaxDiff), withers (WMinDiff, WMaxDiff) and pelvis (PMinDiff, PMaxDiff). A left hind step was defined as the duration from maximum protraction of the left hind to maximum protraction of the right hind, and vice versa for a right hind step. By convention, these differences are calculated such that a positive value indicates right limb asymmetry (higher minimum/lower maximum at midstance and following push-off, *i.e.*, late stance in walk), with head and withers values pertaining to the forelimb and pelvic values to the hind limb. For example, for the pelvis and with the stride starting at left hind maximum protraction (or ground contact), MinDiff is calculated by subtracting the minimum value at the end of the left step from the minimum value at the end of the right step, and MaxDiff is calculated by subtracting the value for the right step from the value for the left step (Fig. 2). Additionally, stride vertical range of motion (ROMz) for the head (HROMz), withers (WROMz) and pelvis (PROMz) were calculated.

Limb variables were ROM for tarsal, stifle and hip joints. Hip joint angle was defined as the global angle between the stifle marker, the hip joint marker and the tuber coxae

marker. Stifle joint angle was defined as the global angle between the tarsal marker, the stifle marker and the hip joint marker. Tarsal joint angle was likewise defined as the global angle between the distal third metatarsal marker, the tarsal marker, and the stifle marker. For each joint, the range of motion (ROM) was the difference between the minimal and maximal angles.

Direction related patterns were evaluated by comparing variable values for left and right circles. To facilitate this, values for left direction were multiplied by (−1) for the following variables: HMaxDiff, WMaxDiff, PMaxDiff, HMinDiff, WMinDiff, PMinDiff, neck-trunk angle and trunk horizontal angle. Following this normalisation to direction, a positive value should be interpreted as follows. A positive MinDiff or MaxDiff indicates inside limb asymmetry with a relatively larger minimum or smaller maximum (following multiplication of left direction values by (−1), the difference values in Fig. 2 would have positive signs). For MinDiff a positive value thus indicates less downward movement when the inside fore (withers) or hind (pelvis) limb is in retracted position and outside limb in protracted position. For the neck-trunk and trunk horizontal angles, positive values indicate displacement of the head or hindquarters to the inside.

## Statistical analysis

Statistical analysis was made using SAS version 9.4. Linear or linear mixed models were developed from stride-level data using the SAS-procedure MIXED.

To address the possible presence of both individual and population level laterality, both horse-specific models and group-level models with data for all horses were made. Horse-specific models were either mixed models (limb variables) or linear models without any random effects (all other variables, *i.e.*, trunk horizontal angle, neck-trunk angle, and trunk vertical motion variables and pelvic rotations, from here on denoted axial body variables). For group-level analyses, mixed models were used. Outcome variables were the biomechanical variables listed above. Fixed effect independent variables in the models with axial body variables as outcome comprised speed, direction, condition and the interaction between direction and condition. Fixed effect independent variables for group-level limb ROM variables comprised speed, condition, direction, and the interaction between direction and inside/outside limb (no interaction was included between direction and condition). Due to limb marker data loss for ridden trials in some horses, individual level limb models were made on data from unridden trials and condition was omitted from fixed effects. Due to incomplete unridden data (due to marker loss), an exception was made for horse Q, for which stifle and hip ROM least square mean (LSM) were based on data from both ridden and unridden conditions. Random effects in group-level models for axial body variables were horse and trial within horse. For the limb joint ROM variables, the random effect was limb nested within trial in the horse-specific models, and horse and limb nested within trial for group-level models.

Group-level models were subsequently modified to address subjective laterality, by adding hollow side as a categorical fixed effect to the group-level model formula described above. Data from two horses were removed from this analysis because riders did not consider them to be hollow in either direction. For axial body variables, the hollow side and
its interaction with direction were the additional fixed effects. For limb variables the added fixed effects were hollow side, the three-way interaction between direction, inside/outside, and hollow side, and all associated two-way interactions.

Before modelling outcome variable distributions (full dataset, all horses) were assessed through inspection of means, medians, skewness, kurtosis and QQ-plotting and Box-Cox transformation (SAS procedure TRANSREG). Transformation was considered for non-normally distributed variables. Residual plots were also evaluated during modelling to ensure adequate normal residual distributions. The general significance level was set to 0.05. Horse-specific models were not reduced, but group-level models were reduced backwards.

K-means cluster analysis was used to investigate agreement between subjective laterality classification (hollow side) and asymmetry patterns in the kinematic variables. LSMs for unridden condition from the horse-specific models were used as input data for this analysis. LSM differences between left and right directions were calculated for each horse and scaled (zscore in Matlab). The scaled left-right differences were then analysed using k-means clustering (kmeans in Matlab, with options 'dist' and 'sqeuclidean'), requesting two groups. All possible sets of three of the 21 outcome variables (1,330 combinations) were evaluated. For each variable set, k-means was run 100 times (kmeans uses a random seed internally). For each run, the cluster group with the largest proportion of left hollow horses was labeled as corresponding to left hollow, and agreement/disagreement with subjectively perceived hollow side was then recorded for each horse. Agreement percentage for each variable set was the calculated by first counting for each horse in how many of the 100 runs that cluster group and subjective categorisation agreed, and then averaging across horses, excluding the horses with hollow side not assigned. The 5% of the variable combinations with highest agreement were extracted. The variables that were included in these 5% top combinations most frequently were tabulated. The method for selecting which cluster group was set to correspond to left and to right hollow yielded a small favour to the left-hollow group, which was slightly larger ($n = 7$) than the right-hollow group ($n = 6$). The cluster analyses were therefore rerun omitting one of the left-hollow horses at a time, to evaluate whether results differed in any appreciable way from those for the full dataset.

## RESULTS

Across variables, the number of strides available for analysis varied between 1,974 and 3,687. The variables with the lowest number of strides with data were WMinDiff (1,974 strides) and WMaxDiff (1,977 strides), for the former there were median 122 strides, with range 82–162 strides per horse. This was due to problems with tracing of the withers marker. All other variables had >2,500 strides (Table S3). In general, the unridden condition had two measurements in each direction and both ridden conditions had one measurement (trial) in each direction for each horse. This means that there was a total of 30 trials for the unridden condition in each direction, but 15 trials per direction each for ridden on long reins and ridden on a contact (short reins). Due to data loss for some markers, there were fewer trials with data for some variables. Circle radius was median
4.3 m, ranging from 3.4 to 4.9 m. Speed ranged from 0.94 to 1.65 m/s, with a median of 1.25 m/s.

Stride data for one horse are plotted by direction and condition in Fig. S1. Most horses, including the one illustrated in Fig. S1, had a greater pelvic pitch stride mean value with a rider regardless of direction, indicating that the pelvis became more horizontal (relatively higher base of the tail).

Table S3 shows descriptive statistics for the variables analysed. Model results for axial body variables can be found in Table 1. Results for hind limb variable models can be found in Table 2. Variable transformations ranged from logarithmic (*e.g.*, tarsal ROM, lambda = 0) to cubed (body tracking angle, lambda = 3). Speed was significant in 13 group-level models (Table S4). Coefficients were negative for neck-trunk angle and HMinDiff indicating that values decreased with increasing speed. The other coefficients were positive, indicating that values increased with speed.

### Group-level differences between conditions

For the axial body variables, the largest difference was between unridden and ridden, whereas differences between long reins and short reins were smaller and not always significant (Table 1). Neck-trunk angle, pelvic pitch ROM, pelvic pitch mean and PROMz showed smaller LSM for the unridden condition. For the other axial body variables, pelvic roll ROM, pelvic yaw ROM, and WROMz, LSM were larger for the unridden condition, as was LSM for tarsal ROM (35.7° *vs.* ≥37°, Table 2).

### Group-level differences between left and right directions

PMaxDiff (left direction −2.8 mm, right direction 5.6 mm), PMinDiff (left −4.9 mm, right direction 4.6 mm) and WMaxDiff (left −3.1 mm, right direction 0.6 mm) all indicated right limb asymmetry in both directions. WMinDiff indicated left (inside) fore asymmetry for left direction (4.2 mm) but was close to zero (0.4 mm) for right direction. When WMinDiff is positive, there is less downward movement during the dual forelimb support with retraction of the inside fore- and protraction of the outside forelimb. HMaxDiff indicated left fore asymmetry for both directions when horses were ridden (left circle 12.6/13.2 mm, right circle −4.3/−5.6 mm for long/short reins) but when unridden a slight inside limb asymmetry was found for both directions (left 1.8 mm, right 4.0 mm). Pelvic roll ROM and pelvic pitch mean were both slightly larger in left direction. Figure 3 illustrates group-model results for axial body parameters relative to direction in a schematic way.

Hip ROM was significantly smaller for the inside limb in both directions, but this was more pronounced going to the left (left circle inside/outside: 16.8° *vs.* 20.7°; right circle: 17.9° *vs.* 19.5°). Stifle ROM was significantly larger for the inside than the outside limb in both directions, but the difference was again more pronounced going to the left (left circle: 43.1° *vs.* 39.0°; right circle: 41.9° *vs.* 40.4°). If taking both directions into account, this suggests that overall, the right hip and the left stifle have larger ROM. Tarsal angle ROM showed no significant effect of direction, but was smaller for the inside limb (inside: 35.6°; outside: 37.6°) (Note that stifle and tarsal motion are functionally linked, so differences

**Table 1 Group-level axial body variables models results.**

| Outcome variable Lambda\|n | Categories | Dir | Cond | LS means Est | SE | BTest | Comparisons Between rows | Type III Effect | p |
|---|---|---|---|---|---|---|---|---|---|
| Neck-to-trunk (°) | Cond | | U | 6.85 | 0.82 | | ● ● | Speed | <0.0001 |
| lambda = 1 | | | L | 13.83 | 1.09 | | ● ○ | Cond | <0.0001 |
| 2,848 | | | S | 15.52 | 1.09 | | ● ○ | | |
| HMaxDiff (mm) | Dir* | L | U | 2,866 | 80.0 | 1.8 | ○ ● ● | Dir | <0.0001 |
| lambda = 1.5 | Cond | L | L | 3,100 | 98.7 | 12.6 | ● ● ○ | Cond | 0.91 |
| 2,815 | | L | S | 3,113 | 98.7 | 13.2 | ○ | Dir* | 0.001 |
| | | R | U | 2,914 | 80.0 | 4.0 | ○ ○ ● | Cond | |
| | | R | L | 2,737 | 98.8 | −4.3 | ● ○ ○ | | |
| | | R | S | 2,710 | 98.7 | −5.6 | ● ○ ● | | |
| WMaxDiff (mm) | Dir | L | | −3.06 | 0.88 | | ● | Dir | <0.0001 |
| lambda = 1\|1,977 | | R | | 0.64 | 0.88 | | ● | | |
| PMaxDiff (mm) | Dir | L | | 958 | 10.8 | −2.8 | ● | Dir | <0.0001 |
| lambda = 1.5\|3,277 | | R | | 1,085 | 10.8 | 5.6 | ● | | |
| WminDiff (mm) | Dir | L | | 5.32 | 0.01 | 4.2 | ● | Dir | <0.0001 |
| lambda = 0\|1,974 | | R | | 5.30 | 0.01 | 0.4 | ● | | |
| PMinDiff (mm) | Dir | L | | −4.90 | 1.17 | | ● | Speed | 0.0003 |
| lambda = 1\|3,277 | | R | | 4.64 | 1.17 | | ● | Dir | <0.0001 |
| Pelvis pitch mean (°) | Dir | L | | 82.59 | 3.31 | | ● | Speed | <0.0001 |
| lambda = 1 | | R | | 82.26 | 3.31 | | ● | Dir | 0.005 |
| 2,589 | Cond | | U | 80.87 | 3.31 | | ● ● | Cond | <0.0001 |
| | | | L | 82.98 | 3.31 | | ● ● | | |
| | | | S | 83.42 | 3.31 | | ● ● | | |
| Pelvis pitch ROM (°) | Cond | | U | 1.98 | 0.03 | 7.25 | ● ● | Speed | <0.0001 |
| lambda = 0\|2,577 | | | L | 2.18 | 0.03 | 8.87 | ● ○ | Cond | <0.0001 |
| | | | S | 2.17 | 0.03 | 8.76 | ● ○ | | |
| Pelvis roll ROM (°) | Dir | L | | 2.19 | 0.06 | 9.0 | ● | Speed | <0.0001 |
| lambda = 0 | | R | | 2.17 | 0.06 | 8.8 | ● | Dir | 0.02 |
| 2,794 | Cond | | U | 2.28 | 0.06 | 9.7 | ● ● | Cond | <0.0001 |
| | | | L | 2.16 | 0.06 | 8.7 | ● ● | | |
| | | | S | 2.10 | 0.06 | 8.2 | ● ● | | |
| Pelvic yaw ROM (°) | Cond | | U | 1.72 | 0.02 | 8.8 | ● ● | Speed | <0.0001 |
| lambda = 0.25 | | | L | 1.69 | 0.02 | 8.2 | ● ○ | Cond | <0.0001 |
| 2,578 | | | S | 1.70 | 0.02 | 8.3 | ● ○ | | |
| WROMz (mm) | Cond | | U | 2.29 | 0.03 | 27.7 | ● ● | Speed | <0.0001 |
| lambda = 0.25 | | | L | 2.17 | 0.03 | 22.3 | ● ○ | Cond | <0.0001 |
| 2,991 | | | S | 2.20 | 0.03 | 23.3 | ● ○ | | |

| Outcome variable Lambda|n | Categories | | LS means | | | Comparisons Between rows | Type III | |
|---|---|---|---|---|---|---|---|---|
| | | Dir | Cond | Est | SE | BTest | | Effect | p |
| PROMz (mm) | Cond | | U | 4.02 | 0.03 | 55.9 | ● ● | Speed | <0.0001 |
| lambda = 0 | | | L | 4.11 | 0.03 | 60.7 | ● ● | Cond | <0.0001 |
| 3,282 | | | S | 4.07 | 0.03 | 58.8 | ● ● | | |

Note:
Shown are least square (LS) means estimates, with back-transformation where relevant (BTest), pairwise comparisons and type III p-values. Independent variables tested are speed, direction (dir), condition (cond) and the interaction direction * condition. Lambda refers to the transformation employed, e.g., lambda 1 translates to no transformation and lambda 0 log transformation. Data are from 15 horses. Circles within a column, demonstrate pairwise comparisons performed between categories: black filled circles indicate comparisons significant at p < 0.05 and open circles are non-significant (p ≥ 0.05). A positive estimate for asymmetry parameters translates to inside limb asymmetry. Est, estimate; SE, standard error; p, p-value. For direction (dir) L, left; R, right; for condition (cond) U, unridden; L long reins; S, short reins.

**Table 2 Group-level models for hip, stifle and tarsal ROM.**

| Outcome variable transform|n | Variable | Categories Dir/ cond | Limb | Least square Means Est | SE | BTest | Between-row Comparisons | Type III Parameter | p-value |
|---|---|---|---|---|---|---|---|---|---|
| Hip ROM | Dir * in/ | Left | In | 16.8 | 0.68 | | ● ● ● | Speed | <0.0001 |
| lambda = 1 | outside | Left | Out | 20.7 | 0.68 | | ● ● ● | In/outside | <0.0001 |
| 5,526 | | Right | In | 17.9 | 0.68 | | ● ● ● | Dir | 0.77 |
| | | Right | Out | 19.5 | 0.68 | | ● ● ● | Dir * in/outside | <0.0001 |
| Stifle ROM | Dir * in/ | Left | In | 1,861 | 61 | 43.1 | ● ● ● | Speed | <0.0001 |
| lambda = 2 | outside | Left | Out | 1,519 | 61 | 39.0 | ● ● ● | In/outside | <0.0001 |
| 5,568 | | Right | In | 1,758 | 61 | 41.9 | ● ● ● | Dir | 0.76 |
| | | Right | Out | 1,635 | 61 | 40.4 | ● ● ● | Dir * in/outside | <0.0001 |
| Tarsal ROM | Condition | U | | 4.91 | 0.01 | 35.5 | ● ● | Speed | <0.0001 |
| lambda = 0 | | L | | 4.92 | 0.01 | 37.0 | ● ○ | Condition | <0.0001 |
| 5,568 | | S | | 4.92 | 0.01 | 37.3 | ● ○ | In/outside | <0.0001 |
| | In/ | | In | 4.91 | 0.01 | 35.6 | ● | | |
| | outside | | Out | 4.92 | 0.01 | 37.6 | ● | | |

Note:
Models are based on 15 horses walking on left and right circles (left and right directions) in three conditions (unridden (U), and ridden on long (L) or short reins (S)), and compares between directions and between if the limb is an inside (In) or outside (Out) limb. The lambda used for transformation and the number (n) of observations (strides) are shown in the first column. Circles demonstrate pairwise 'between' 'row' comparisons performed: black filled circles indicate comparisons significant at p < 0.05 and open circles non-significant, i.e., p ≥ 0.05. Est, estimate; SE, standard error; BTest, back-transformed estimate.

between the values reported here are functions of the statistical models, rather than a linked opposite effect between those motions).

## Subjectively perceived laterality

Participating riders were asked questions related to their perception of the horse's stiff and hollow side (Table S2) and most questions were answered. Based on these questionnaire data (Tables S1 and S2), seven horses were categorised as hollow left and six as hollow right. Two horses were said to be equal on the two sides ('neither side') and were eliminated from the models that included subjective laterality as a variable. Agreement

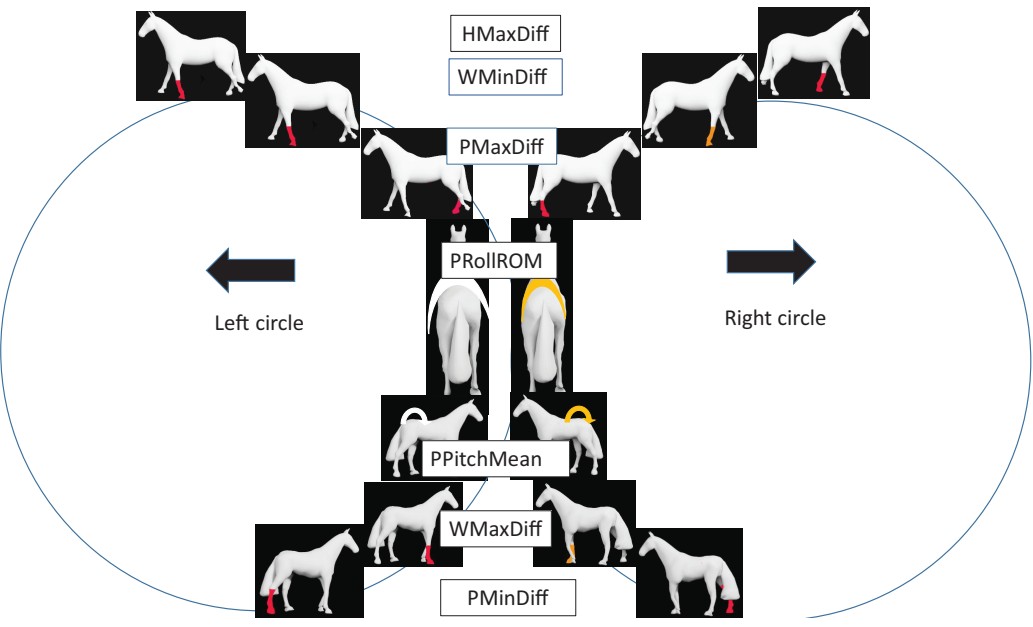

**Figure 3 Schematic presentation of group model results related to direction for horses walking on circles, ignoring hollow side (Table 1).** Each pair of horses aligned horizontally shows asymmetries found between left and right circles. For vertical movement asymmetry parameters, coloured limbs are shown as fore/hind, left/right, inside/outside and whether they represent midstance or endstance. Limb colour demonstrates least square mean asymmetry: RED > 1 mm, ORANGE ≤ 1 mm. WHITE arrows indicate greater movement than YELLOW arrows, *i.e.*, more pelvic roll range of motion (PRollROM) on the left circle and increased pelvic pitch mean (PPitchMean) with a more horizontal pelvis indicating increased extension (base of tail raised) on the left circle. Head maximum difference (HMaxDiff) results are only relevant for the ridden conditions. WMaxDiff, withers maximum difference; PMaxDiff, pelvic maximum difference; WMinDiff, withers minimum difference; PMinDiff, pelvic minimum difference.

between the answers to individual questions and overall categorisation of the horses as left hollow, right hollow, or neither is shown in the right-most column in Table S2 (disagreement is indicated with zeros, four occurrences; one indicates agreement, 31 occurrences).

When subjective laterality, represented by rider perceived hollow side, and its interaction with direction, were added to the axial body variable group-level models (Table 3), the interaction was significant for trunk horizontal angle ($p \leq 0.05$), PMaxDiff ($p = 0.006$), HMinDiff ($p = 0.006$) and pelvic roll ROM ($p = 0.003$). For hind limb joint ROM variables, the three-way interaction between direction, inside/outside limb, and hollow side was significant for hip ($p < 0.0001$) and tarsal ($p = 0.003$) joint ROM (Table S5). After removing either horse C or H, the horses for which hollow side was least clear from the riders' answers, all significances remained except for trunk horizontal angle when horse C was removed (essentially borderline also before removal).

Trunk horizontal angle tended to be slightly more negative in the right direction for right-hollow ($-0.7°$) compared to left-hollow horses ($-0.1°$) (Table 3). This suggests a tendency for right-hollow horses to move with the hindquarters slightly to the outside in the right direction (pairwise comparison $p = 0.07$). For PMaxDiff, the consistent right hind

**Table 3 Group-level axial body variable models including hollow side as a variable (left L/right R).**

| Outcome Variable\|n | Variable categories Dir/cond | Hollo | LS means Est | SE | BTest | Between-row Comparisons | | | Type III Effect | p-value |
|---|---|---|---|---|---|---|---|---|---|---|
| Trunk horizontal angle 2,624 | Dir* | L | 991,069 | 6,762 | −0.3 | ○ | | ○ | Dir | 0.49 |
| | Hollo L | R | 992,462 | 7,310 | −0.3 | ○ | | ○ | Hollo | 0.30 |
| | R | L | 997,741 | 6,766 | −0.1 | ○ | ○ | | Dir* | 0.05 |
| | R | R | 978,631 | 7,303 | −0.7 | ○ | | ○ | Hollow | |
| PMaxDiff 2,885 | Dir* | L | 941 | 16 | −3.9 | ○ | | ● | Dir | <0.0001 |
| | Hollo L | R | 981 | 18 | −1.3 | ○ | | ● | Hollo | 0.46 |
| | R | L | 1,093 | 16 | 6.1 | ○ | ● | | Dir* | 0.01 |
| | R | R | 1,088 | 18 | 5.8 | ○ | | ● | Hollo | |
| HMinDiff 2,456 | Dir* | L | −7.41 | 5.58 | | ○ | | ○ | Dir | 0.11 |
| | Hollo L | R | 0.21 | 6.04 | | ○ | | ● | Hollo | 0.88 |
| | R | L | −3.65 | 5.59 | | ○ | ○ | | Dir* | 0.01 |
| | R | R | −13.52 | 6.04 | | ○ | | ● | Hollo | |
| Pelvic roll ROM 2,416 | Cond | U | 2.28 | 0.07 | 9.8 | ● | ● | | Speed | <0.0001 |
| | | L | 2.17 | 0.07 | 8.7 | ● | | ● | Dir | 0.07 |
| | | S | 2.11 | 0.07 | 8.3 | | ● | ● | Cond | <0.0001 |
| | Dir* | L L | 2.16 | 0.09 | 8.7 | ○ | | ● | Hollo | 0.42 |
| | Hollo | L R | 2.23 | 0.10 | 9.3 | ○ | | ○ | Dir* | 0.003 |
| | | R L | 2.11 | 0.09 | 8.2 | ○ | ● | | Hollo | |
| | | R R | 2.25 | 0.10 | 9.5 | ○ | | ○ | | |

**Notes:**
Shown are least square (LS) means, with back-transformation (BTest) where necessary, pairwise comparisons and type III p-values. For transformations see Table 1, except for trunk horizontal angle where lambda = 3. Data are from seven left and six right-hollow horses, as evaluated from the riders' answers (Table S2). Circles demonstrate pairwise comparisons performed: black filled circles indicate comparisons significant at $p < 0.05$ and open circles non-significant $p \geq 0.05$ (the non-significant comparison for right circle between left and right-hollow horses is associated with $p = 0.07$). A positive estimate for vertical motion asymmetry parameters translates to inside limb asymmetry. The number (n) of observations (strides) are shown in the first column.
Est, estimate; SE, standard error; p, p-value; Dir, direction; L, left; Hollo, hollow; R, right; Cond, condition; U, unridden; L, long reins; S, short reins.

asymmetry found in the group-level model without subjective laterality (Table 1) was numerically more obvious for left-hollow horses (left direction −4 mm, right direction 6 mm) than for right-hollow horses (left −1 mm, right 6 mm), though values for left- vs. right-hollow horses did not differ significantly in either direction. Left-hollow horses had greater pelvic roll ROM moving in left (8.7°) vs. right (8.2°) direction ($p = 0.0005$), similar to the group-level result. Again similar to the group-level model without subjective laterality (Table 2), left-hollow horses had smaller inside and greater outside hip joint ROM in left (inside 17.0°, outside 21.9°) vs. right direction (inside 18.9°, outside 20.1°, both $p < 0.0001$). Both of these comparisons were non-significant for right-hollow horses, neither pelvic roll nor hip ROM differed between left and right directions. Right-hollow horses instead had a significant difference in HMinDiff between left (0 mm) and right (−14 mm) directions, indicating less lowering of the head at midstance of the outside forelimb in right direction (hollow side as inside), and larger outside tarsal ROM in right (38.6°) vs. left (37.4°) direction ($p \leq 0.05$). Figures 4 and 5 illustrates group-model results for left-hollow and right-hollow horses schematically.

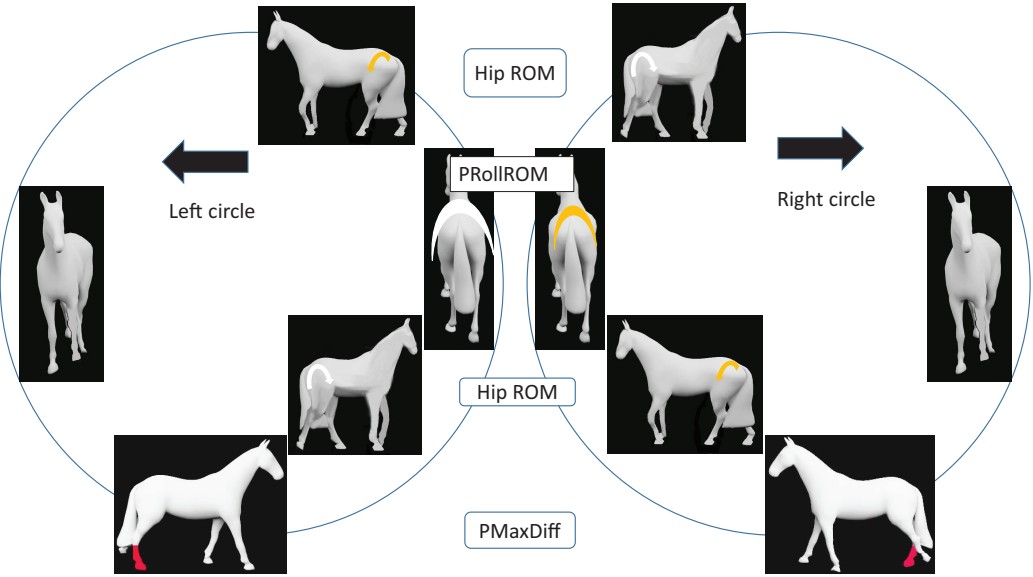

**Figure 4 Results for left hollow horses walking on circles (Tables 3, S5).** Coloured symbols show asymmetries between LEFT and RIGHT circles. For vertical movement asymmetry parameters, coloured limbs are shown as fore/hind, left/right, inside/outside and whether at midstance or endstance. RED limb colour indicates asymmetry >1 mm. WHITE limb colour indicates greater pelvic roll range of motion (PRollROM) or increased hip ROM, compared to YELLOW. The four findings illustrated show inside hip ROM: RIGHT > LEFT; outside hip ROM: LEFT > RIGHT; pelvis roll ROM: LEFT circle > RIGHT circle. Pelvis vertical maximum at right hind midstance is relatively lower on both circles. Horses without coloured symbols are only included for visualization of how left-hollow horses may often be described by equestrians. PMaxDiff, pelvic maximum difference.               

## Horse-specific models

Results for direction in the horse-specific models are summarised in Table 4. Each column represents one horse, and the rows indicate variables. Within a cell, results for the between-directions comparison for each condition (unridden, ridden on long reins, and ridden on short reins, respectively, in that order) are indicated with L (significantly larger value in left direction), R (significantly larger value in right direction), or—(no significant difference). For example, horse B has three Ls on the row for trunk horizontal angle, which indicates that the hindquarters were more towards the inside/less towards the outside of the circle moving in left vs. right direction for all conditions. Note that results for vertical motion asymmetry parameters have not been illustrated, because of difficulties in presenting these in a manner comparable to that of the other parameters (Table 4). Only data from unridden trials were evaluated in the limb models, hence only one LSM is presented per direction and inside/outside limb.

## K-means clustering

K-means clustering was performed with the number of cluster groups set to two. All possible sets of three of the kinematic variables were evaluated as input. Agreement percentage between subjective laterality and cluster groups ranged between 50% (no better than chance) and 80.5%, with median agreement 62%. For the 5% sets with the highest

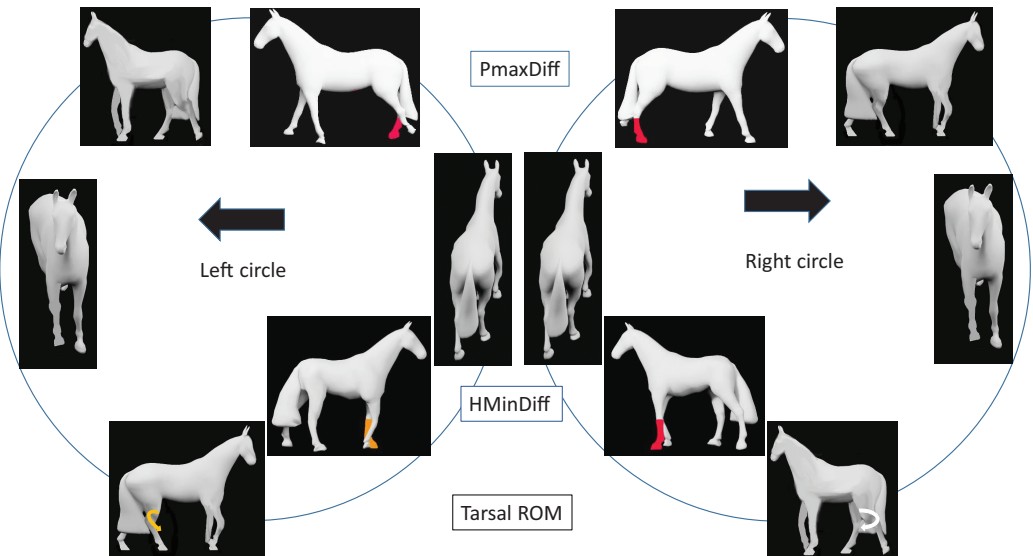

**Figure 5 Results for right hollow horses walking on circles (Tables 3, S5).** Coloured symbols show asymmetries between LEFT and RIGHT circles. For vertical movement asymmetry parameters, coloured limbs are shown as fore/hind, left/right, inside/outside and whether at midstance or endstance. RED limbs have least square mean asymmetry >1 mm, and ORANGE limbs ≤ 1 mm. For tarsal range of motion (ROM), WHITE arrows > YELLOW arrows. Illustrated findings show that outside tarsal ROM is larger on the RIGHT circle, HMinDiff shows left forelimb midstance asymmetry on both circles and PMaxDiff shows right hind limb asymmetry on both circles. Horses without coloured symbols are only included for visualization of how right-hollow horses are described by equestrians. PMaxDiff, pelvic maximum difference; HMinDiff, head minimum difference.

agreement (67 sets of 1,330 evaluated), agreement ranged between 71.2% to 80.5% . The variable set with highest agreement (80.5%) comprised pelvic roll ROM, pelvic pitch ROM and outside limb tarsal ROM. Table 5 lists the 10 sets with the highest agreement and in Table S6 all 67 top combinations are shown. The variable that appeared most frequently in the top 5% sets was pelvic roll ROM, followed by HMinDiff and PMaxDiff (Table 6). Omitting one of the left-hollow horses at a time yielded similar results (Table S6) regarding which variables were most frequent in sets with high agreement. Pelvic roll ROM and HMinDiff were still the most frequently included variables in the top 5% sets, but the third most frequent variable varied depending on which horse was removed.

## DISCUSSION

The studied group of 15 horses, of varying size and breed, moved significantly differently when walking on circles in left *vs.* right directions. The right hip and the left stifle had relatively larger ROM in both directions. Further, comparing left and right sides, the pelvis consistently reached a higher maximum position at left hind midstance and a lower minimum position in terminal left hind stance, which overlaps with early right hind stance. The latter finding suggests that the horses retracted the left hind limb further than the right hind limb, which would result in a larger inter-limb distance at this moment.

**Table 4 Result for direction in the horse-specific models (not including results for vertical motion asymmetry parameters).**

| Angles/distances | Horse | | | | | | | | | | | | | | |
|---|---|---|---|---|---|---|---|---|---|---|---|---|---|---|---|
| | B | C | D | H | J | V | Y | A | Q | F | I | M | P | S | X |
| Trunk horizontal (°) | L-L-L- | ------ | -L-L- | -L-L- | R-R-R- | ------ | --L- | ------ | ------ | R----- | -L-L- | ------ | ------ | L-R-R- | -L-L- |
| Neck-to-trunk (°) | ------ | ------ | -L-L- | ------ | L-R-R- | ------ | L-L-L- | R-R-R- | ------ | L----- | -L-L- | L-L-L- | -L-L- | R-R-L- | R-R-L- |
| Pelvis pitch mean (°) | L-L-L- | ------ | ------ | ------ | ------ | ------ | L-L-- | ------ | L-L-L- | ------ | R-R-R- | ------ | L-L-L- | R--R- | ------ |
| Pelvis pitch ROM (°) | ------ | ------ | ------ | ------ | ------ | ------ | ------ | ------ | L--L- | ------ | ------ | ------ | ------ | ------ | ------ |
| Pelvis roll ROM (°) | L----- | ------ | ------ | ------ | ------ | ------ | ------ | ------ | ------ | ------ | ------ | ------ | --L- | ------ | ------ |
| Pelvis yaw ROM (°) | ------ | ------ | ------ | ------ | ------ | ------ | ------ | ------ | ------ | ------ | R-R-- | ------ | -L-- | ------ | ------ |
| HROMv (mm) | ------ | ------ | ------ | ------ | -L-- | ------ | --L- | ------ | -L-- | ------ | ------ | -L-L- | ------ | -L-- | R-R-R- |
| WROMv (mm) | ------ | ------ | -L-L- | ------ | ------ | ------ | L-R- | L-R-L- | L-L- | ------ | ------ | -L-- | ------ | ------ | -R-- |
| PROMv (mm) | -R-R- | ------ | L----- | ------ | ------ | L---- | ------ | ------ | -R-L- | ------ | L----- | ------ | R-L-R- | ------ | L-L--- |
| Hip Inside ROM (°) | R | | R | R | R | R | R | R | R | | L | R | R | | L |
| Hip outside ROM (°) | R | | | R | | R | | L | | | R | L | | | R |
| Stifle inside ROM (°) | L | L | L | L | R | L | L | | | | | L | | L | L |
| Stifle outside ROM (°) | | | | L | L | L | | | | | | L | L | | |
| Tarsal inside ROM (°) | | L | | | R | L | R | L | | R | | L | | | L |
| Tarsal outside ROM (°) | | R | | | L | L | L | R | R | L | R | R | | R | R |
| Hollow side | L | L | L | L | L | L | L | 0 | 0 | R | R | R | R | R | R |

**Note:**
For the remaining axial body variables there are three possibly significant ($p < 0.05$) results for each variable, from left to right within the cell these refer to the unridden condition, ridden on long reins, and ridden on short reins. When results are significant, the letter L (left) or R (right) shows the direction with the larger LSM estimate. For limb variables only data from the unridden condition were included. Hollow side from subjective evaluation is included at bottom of table (0 = undetermined/'neither' side). HROMz, head vertical range of motion; WROMz, withers vertical range of motion; PROMz, pelvis vertical range of motion.

**Table 5 The 10 three-variable combinations with the highest agreement with hollow side (based on data from Table S2).**

| Variables in the combination | | PRollROM | | | | | | | | | |
|---|---|---|---|---|---|---|---|---|---|---|---|
| Hollow | Horse | HMinDiff HipIns | HMinDiff PYawROM | TarsOut HipIns | PPitchROM TarsOut | HMinDiff HipOut | HMinDiff PMaxDiff | WMinDiff TarsOut | PPitchMean TarsOut | HipOut TarsOut | HMinDiff TarsOut |
| Left | B | **100** | 98 | 99 | 55 | **100** | **100** | 66 | **93** | 92 | 93 |
| Left | C | **100** | 97 | 26 | 47 | **95** | 70 | 36 | 18 | 36 | 60 |
| Left | D | **100** | 40 | 99 | 77 | **100** | 26 | 65 | **95** | 94 | 93 |
| Left | H | **100** | 98 | 99 | 97 | **100** | **100** | 70 | **95** | 97 | 93 |
| Left | J. | **100** | 98 | 99 | 97 | **100** | **100** | 68 | **95** | 98 | 93 |
| Left | V | **100** | 65 | 97 | 94 | **100** | **100** | 65 | 87 | 92 | 100 |
| Left | Y | 19 | 4 | 31 | **96** | 16 | 13 | 59 | **84** | 54 | 26 |
| Neither | A | 100 | 98 | 27 | 12 | 100 | 99 | 29 | 17 | 61 | 55 |
| Neither | Q | 100 | 69 | 99 | 96 | 100 | 26 | 66 | 94 | 99 | 93 |
| Right | F | 55 | **89** | 69 | 48 | 46 | **89** | **97** | 84 | 61 | 39 |
| Right | I | **89** | **96** | 88 | 50 | **84** | **87** | **98** | 86 | 81 | 82 |
| Right | M | 67 | **96** | 79 | 90 | 59 | **87** | **100** | 88 | 77 | 82 |
| Right | P | **85** | **96** | 76 | 93 | **90** | **87** | **99** | 60 | 74 | 82 |
| Right | S | 49 | 66 | 78 | 95 | 48 | 74 | **99** | 92 | 74 | 83 |
| Right | X | **89** | **96** | 88 | 88 | **84** | **87** | **98** | 40 | 81 | 82 |
| No. runs agreement left | | 619 | 500 | 550 | 563 | 611 | 509 | 429 | 567 | 563 | 558 |
| No. runs agreement right | | 434 | 539 | 478 | 464 | 411 | 511 | 591 | 450 | 448 | 450 |
| No. agreement | | 1,053 | 1,039 | 1,028 | 1,027 | 1,022 | 1,020 | 1,020 | 1,017 | 1,011 | 1,008 |
| Agreement over runs (%) | | 81.0 | 79.9 | 79.1 | 79.0 | 78.6 | 78.5 | 78.5 | 78.2 | 77.8 | 77.5 |

Note:
The results were derived running k-means clustering 100 times for each of the 1,330 three-variable combinations evaluated from 21 variables on 15 horses. Bold numbers have over 75% agreement for left and right hollow horses, respectively. For the horses without sidedness (Neither) a high number suggests they are left hollow and a low number right hollow. Pelvic roll range of motion (ROM) participates in all combinations. Agreement is calculated with 1,300 (13 horses with left or right hollow side* 100 runs) as denominator. PRollROM, pelvic roll ROM; PPitchROM, pelvic pitch ROM; HMinDiff, head minimum vertical difference; HipIns, inside hip angle ROM; HipOut, outside hip angle ROM; WMinDiff, withers minimum vertical difference; TarsOut, outside tarsal angle ROM; PMaxDiff, pelvis maximum vertical difference; PYawROM, pelvic yaw ROM; PPitchMean, pelvic pitch mean.

These hind limb asymmetries follow the expected pattern for walking on a right circle (*Egenvall, Engström & Byström, 2020*). However, the horses in the current study showed this pattern regardless of direction. Perhaps the horses adapted better to, or were better balanced on the right circle, as shown by their ability to maintain more symmetric hip and stifle ROM and withers vertical motion in the right direction.

Overall, the kinematic asymmetries identified between left and right directions for the horses as a group did not agree directly with the riders' impressions of the horses' hollow and stiff sides (subjectively perceived laterality), even if there was some overlap. The number of horses perceived as left and right lateralised, respectively, was relatively similar, whereas the significant effects of direction at group-level suggest a population bias. This is in accordance with a previous study that compared different methods to determine horses' laterality (*Kuhnke & König von Borstel, 2022*), where it was found that the results of different laterality tests generally did not agree. This suggests that laterality can manifest in

**Table 6 The variables appearing most often in the 5th percentile highest agreement with subjective laterality.**

| Variable | Count | Percent |
|---|---|---|
| Pelvic Roll ROM (°) | 102 | 25.6 |
| HMinDiff (mm) | 48 | 12.0 |
| PMaxDiff (mm) | 37 | 9.3 |
| Hip inside ROM (°) | 29 | 7.3 |
| Hip outside ROM (°) | 25 | 6.3 |
| Tarsal outside Rom (°) | 21 | 5.3 |
| Pelvic pitch ROM (°) | 19 | 4.8 |
| Tarsal inside ROM (°) | 12 | 3.0 |
| Trunk horizontal (°) | 12 | 3.0 |
| Pelvic yaw ROM (°) | 11 | 2.8 |
| Neck-trunk (°) | 11 | 2.8 |
| Stifle inside ROM (°) | 9 | 2.3 |
| PROMz (mm) | 9 | 2.3 |
| HROMz (mm) | 9 | 2.3 |
| WMinDiff (mm) | 9 | 2.3 |
| Pelvic pitch mean (°) | 8 | 2.0 |
| Stifle outside ROM (°) | 6 | 1.5 |
| WMaxDiff (mm) | 6 | 1.5 |
| HMaxDiff (mm) | 6 | 1.5 |
| WROMz (mm) | 5 | 1.3 |
| PMinDiff (mm) | 5 | 1.3 |

**Note:**
The results were derived running k-means clustering 100 times for each of the 1,330 three-variable combinations evaluated from 21 variables on 15 horses (and calculated on 67 three-variable combinations). HMaxDiff, head maximum vertical difference; WMaxDiff, withers maximum vertical difference; PMaxDiff, pelvic maximum vertical difference; HMinDiff, head minimum vertical difference; WMinDiff, withers minimum vertical difference; PMinDiff, pelvis minimum vertical difference; HROMz, head vertical range of motion; WROMz, withers vertical range of motion; PROMz, pelvis vertical range of motion.

different ways that may not be related to each other. Subjective laterality was significant for three of nine variables with a group-level effect of direction. For all three, the group-level differences between directions were either significant only for L-hollow horses (pelvic roll ROM, hip ROM), or they were numerically larger in that subgroup (PMaxDiff).

For left-hollow horses, pelvic roll ROM was slightly larger for the left direction, which was not the case for right-hollow horses. This may relate to why the left-hollow horses were perceived to be stiffer to the right. However, riders do not necessarily perceive the hollow side as the better side. In fact, the stiff side may well be more stable, while on the hollow side the movements seem overly mobile in a non-functional way. Right-hollow horses lowered the head relatively less at outside forelimb midstance and showed greater outside tarsal ROM when moving on a right *vs.* left circle, which was also found for the whole group. This may imply that left and right hollow horses are not mirror images but show fundamental differences. A corresponding conclusion has previously been advocated for handedness in humans (*Schott & Schott, 2004*).

## Head and withers motion

During walk in a straight line, the head reaches its lowest position close to forelimb midstance and its highest position in late forelimb stance (*Loscher et al., 2016*; *Rhodin et al., 2022*). HMinDiff was associated with forelimb lameness in unridden horses walking in a straight line, in an induced lameness model (*Serra Bragança et al., 2021*). In the lame forelimb there was concurrently an attenuation of the second vertical ground reaction force peak, which occurs just after forelimb midstance. In the current study, HMinDiff indicated outside forelimb asymmetry for right-hollow horses when walking on a right circle (−14 mm). If the relation between head movement and limb loading is similar for lameness and for normal walk on a circle, this would suggest that right-hollow horses had decreased weight-bearing on the left (outside) forelimb in right direction. In trot, offloading of a forelimb will result in both head and withers vertical motion asymmetry (*Persson-Sjödin et al., 2023*). However, neither WMinDiff nor WMaxDiff have been found to be associated with lameness in walk, at least not on a straight line (*Buchner et al., 1996*; *Serra Bragança et al., 2021*). This may reflect the fact that head and withers vertical movements in walk are interconnected in a different manner, compared to trot (*Loscher et al., 2016*). In horses walking on a treadmill, WMinDiff, but not WMaxDiff, has been suggested to be related to laterality (*Byström et al., 2018*). None of the withers vertical motion variables showed any significant association with hollow side in the current study. However, there was a group-level effect of direction for WMinDiff indicating relatively less downward movement during early right fore, late left fore stance in the left direction, and concurrently HMaxDiff indicated group-level left fore asymmetry for both directions, the latter finding only when the horses were ridden. Similar to WMinDiff and WMaxDiff, HMaxDiff does not appear to be associated with lameness in walk (*Serra Bragança et al., 2021*), which makes it more likely that these findings reflect laterality, even though this pattern does not appear to be analogous to the riders' perception of the horse having a stiff and hollow side.

## Horse-specific models and individual variation

In the horse-specific models presented, the condition * direction interaction was frequently significant for trunk horizontal angle and neck-trunk angle (see upper part of Table 4), suggesting individual-level asymmetry for these variables. On the other hand, in the group models, direction was non-significant for both these variables (Table 1), indicating that there was no consistent group-level bias. In the cluster analysis, axial body ROM parameters had relatively few significant differences between directions within condition in the horse-specific models, as well as limited significant results in the group models. Horse-specific results for limb ROM variables had similarities to group level models for inside hip ROM (right inside larger than left inside for nine horses) and inside stifle ROM (left inside larger than right inside for seven horses), while other limb ROM results are relatively less similar. The potential usefulness of horse-specific mixed modelling in equine biomechanics has yet to be explored. Perhaps horse-specific modelling could aid in the evaluation of equine laterality, if we learn how to measure and interpret results from various asymmetry variables.

## Subjective laterality

Some equestrian literature suggests that horses show population-level laterality and that left-hollow horses are more common than right-hollow horses (*e.g.*, *Von Ziegner, 2002*). The many differences between left and right direction in group-models support the notion that population-level laterality exists in horses. However, approximately 50% of the horses were subjectively categorised as left ($n = 7$) or right ($n = 6$) lateralised, with two horses perceived as not having a preference. In classifying their horses, the opinions of individual riders may have been influenced by their training or by peers. However, subjective laterality designation is still essential in order to study laterality as found in real life, even if for example the rider's own laterality or previous injuries may confound answers to an unknown extent. In addition to the effects of lameness and laterality, random left-right asymmetries may arise due to differences in strength or timing of the signals from the central pattern generators in the spinal cord (*e.g.*, *Kuhtz-Buschbeck et al., 2008*). Thus, it is possible that several compound patterns exist (c.f. *Kuhnke & König von Borstel, 2022*), and that different equestrians have focused on different aspects. In the current study, both circle direction and rider perceived laterality were associated with biomechanical asymmetry patterns, possibly suggesting that the study horses displayed two different kinds of patterns. However, as horses were few, incorrect designation for hollow side, under the presumption that there is a true correct but unknown status, may have a large influence on the analysis given the small number of horses included.

## Ridden *vs.* unridden

In general, horses showed positive neck-trunk angles indicating that the head is usually carried to the inside of the circle in walk, although in occasional strides the head was to the outside (Table S3, Fig. S1). The neck-trunk angle was larger, *i.e.*, the horses kept the head more to the inside, with a rider (LSM: unridden: 7°; ridden with short reins: 16°). At the same time, HMaxDiff indicated left fore asymmetry for both directions when horses were ridden, but when unridden a slight inside limb asymmetry was found for both directions (left 1.8 mm, right 4.0 mm). It is possible that it was easier for the horses to achieve symmetric vertical head movements between directions with less bending, but it may also be related to the rider. Withers vertical excursion (WROMz) was smaller when ridden compared to unridden both in the current and in a previous study (*Egenvall, Engström & Byström, 2020*), suggesting a mechanical effect of the addition of the rider's weight. In fact, most ROM variables with significant differences showed smaller values with a rider, except pelvic pitch ROM and PROMz. A couple of previous studies also suggest that asymmetry may increase with a rider (*Peham et al., 2004*; *Byström et al., 2021*), in spite of the fact that achieving straightness is a cornerstone in dressage training (*Fédération Equestre Internationale (FEI), 2022*). One reason for the consistent head motion asymmetry in both directions could be that all riders in the study were right-handed which is associated with stronger tension in the left rein (*Kuhnke et al., 2010*). Pelvic pitch mean was larger, indicating more extension when horses were ridden, which may reflect the effect of the rider's weight (*De Cocq, Van Weeren & Back, 2004*).

## Combined analysis

Associations between subjectively perceived laterality and biomechanical variables were further investigated using k-means clustering. K-means clustering analysis allows evaluation of several candidate variables together in sets, rather than in separate models. However, it does not evaluate their functional relationship and the results offer no biological rationale for how asymmetries may interact. Agreement between subjectively perceived hollow side and cluster groups ranged between 50–80%, which seems to overrate the actual agreement from the fact that agreement by chance is not taken onto account. In contrast, in Kappa analysis of agreement, estimates are adjusted for agreement due to chance (*Cohen, 1960*). No such correction was attempted in the current study, since the primary use of these figures was relative comparisons between variable sets. Further, since agreement in the top-ranked combination was only slightly better than that in the next-highest ranked combination, we deemed it more relevant to look at how many times each variable was included in the 5% sets with the highest agreement, rather than drawing conclusions from the top combination alone. On this basis, pelvic roll ROM was found to be the most influential variable for determining subjective laterality, with several other variables also being important (*e.g.*, HMinDiff, PMaxDiff).

In the 10 sets with the highest agreement (Table 5) some horses were classified consistently across both sets and runs (*e.g.*, horse H, all sets strongly suggest left hollow). For other horses classification was more ambiguous (*e.g.*, horse C). For a few horses, the cluster classification disagreed more or less consistently with the subjective evaluation, suggesting these horses are somehow dissimilar to the other horses subjectively perceived as hollow to the same side. For example, horse Y, subjectively classified as left-hollow, had agreement below 50% for most of the 10 sets. Accordingly, the data suggested that this horse was most likely right-hollow. The horses deemed subjectively to not show a side preference (horses A and Q) also appeared ambiguous in the clustering results. Horse A was categorised as right-sided four times and as left-hollow six times. Horse Q was most often classified as left-hollow in the 10 sets with highest agreement. It would be interesting to explore this approach in a larger group of horses, and preferably include subjective assessment by multiple riders, to further elucidate the usefulness of cluster analysis as a means of identifying laterality-related patterns in horses.

## Benefits and limitations

The number of horses was smaller than ideal, given that the objective was to study asymmetries (*Egenvall, Marr & Byström, 2021*). Horses also differed widely in age and breed. While it may be debated among equestrians whether horses are more or less asymmetric when young or old, older horses may be more likely to have acquired injuries associated with lameness. Achieving straightness is one of the goals in riding, suggesting that horses are more crooked when having been less trained and thus that younger horses may be more asymmetric, even if evidence of increasing asymmetry with age comparing foals and 2-year olds has been found (*Lucidi et al., 2013*). Also, *McGreevy & Thomsen (2006)* found, studying whether the left or right forelimb is most often advanced while grazing, that individual-level bias became stronger with age, while they found no evidence

for increasing population-level bias with age. The same study also found breed-differences in motor laterality (*McGreevy & Thomsen, 2006*). Only if laterality is expressed in fundamentally different ways, and not just differ in degree, by age and breed would this be a problem for the current analyses. However, this question deserves attention and breed and age (and gender) effects could be investigated in a larger study in the future.

The two major challenges in studies of laterality are to verify that the included horses are sound and to verify laterality subjectively. Asymmetries at trot in horses perceived as sound by the owners, as well as by experienced equine clinicians, often exceed thresholds for low-grade lameness (*Rhodin et al., 2017*; *Hardeman et al., 2022*), and it is currently impossible to distinguish between these groups based on the measurements alone. Arguments for studies of laterality in walk include that low-grade lameness likely has a smaller influence on motion symmetry in walk compared to trot, and that the impact from laterality is possibly larger in walk than in trot (*Byström et al., 2018*).

Laterality was indeed a subjective variable (Table S2). Given the low number of horses, results related to laterality will be sensitive to 'erroneous' classification. Neither behaviour-related scoring (as for example done by *Schwarz et al., 2022*) or scrutinisation of fore hoof conformation was made (*Van Heel et al., 2006*). Another major challenge was to relate biomechanical parameters with equestrian perceptions of laterality, *e.g.*, what biomechanical variable would reflect lateral deviation of a shoulder or hind limb relative to the general direction of motion. In this aspect, we may not have achieved a perfect match between what we measured and what the riders were describing. A further problem when studying asymmetry is to place skin markers symmetrically, which is required in order to register small differences between the two sides. Mean values are especially sensitive to erroneous marker placement while ROM values are considered more robust (*Audigié et al., 1998*), and when selecting variables for the current analyses care was taken to only include those in which marker placement would have a limited effect.

For individual-level models, limb variables were analysed using data from unridden trials only, since some horses did not have complete limb data for all conditions due to loss of markers. Also, subjective evaluation of laterality was done on horse level. The power for this variable was lower than for measurements that can vary, for example, within a trial. The number of horses was determined by availability and there was no power calculation behind the size of the study group.

In the k-means cluster analysis there is no guarantee that the corresponding cluster group is allocated to the same cluster number across repeated runs, and the group sizes are also free to vary, with two cluster groups between 1 and n−1. To allow comparison to subjective laterality, it was necessary to formulate criteria for labelling the cluster groups as belonging to the hollow or stiff side, and the choice of criteria may influence the outcome of the analysis. As we were unsure how much bias the slightly differently sized laterality groups created, a sensitivity analysis was deemed warranted. Re-running the analysis while excluding one horse at a time yielded similar results to the full analysis. This indicates that the criteria used produced stable results in this respect, but should be (re)examined if using this method on groups with more unequal sizes.

## CONCLUSIONS

Population differences between horses walking in left and right directions were found for several variables, at both group and individual level, together with evidence of associations between biomechanical asymmetries and subjectively assigned laterality. The horses adapted better to, or were better balanced on the right circle, since they maintained more symmetric hip and stifle ROM and withers vertical motion when walking in the right compared to the left direction. Findings suggest that left and right lateralised horses may not be perfect mirror images. Pelvic roll ROM in walk emerged as a promising variable to determine laterality as perceived by the rider, especially when considered together with other variables. However, as in many studies of asymmetry, the cause of the asymmetries found cannot be definitively identified and underlying pathology could not be ruled out entirely. The methods and findings are suggested as a step forward in elucidation of locomotor laterality in horses. For the future, we suggest that this methodology be repeated on more horses and in other gaits, as well as repeatedly on the same horses during their lifetime, to explore further the associations between variables. Additional parameters, such as limb placement relative to the body, should also be measured.

## ACKNOWLEDGEMENTS

We thank the horse owners that volunteered participation with their horses and Hanna Engström for providing access to horses and facilities.

### Funding

Funding for the experiment was provided by a Career Grant provided to Agneta Egenvall from the Swedish University of Agricultural Sciences 2017. The funders had no role in study design, data collection and analysis, decision to publish, or preparation of the manuscript.

### Grant Disclosures

The following grant information was disclosed by the authors:
Swedish University of Agricultural Sciences 2017.

### Competing Interests

The authors declare that they have no competing interests.

### Author Contributions

- Agneta Egenvall conceived and designed the experiments, performed the experiments, analyzed the data, prepared figures and/or tables, authored or reviewed drafts of the article, and approved the final draft.
- Hilary M. Clayton analyzed the data, authored or reviewed drafts of the article, and approved the final draft.

- Anna Byström analyzed the data, authored or reviewed drafts of the article, and approved the final draft.

## Human Ethics

The following information was supplied relating to ethical approvals (*i.e.*, approving body and any reference numbers):

No institution approved the study as this is not needed according to Swedish law.

## Data Availability

The data and codes are available in the Supplemental Files and at figshare: egenvall, agneta; Byström, Anna; Clayton, Hilary M. (2023). Pilot study of locomotor asymmetry in horses walking in circles with and without a rider. figshare. Dataset. https://doi.org/10.6084/m9.figshare.23646363.v1.

## Supplemental Information

Supplemental information for this article can be found online at http://dx.doi.org/10.7717/peerj.16373#supplemental-information.

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
