# Peer review of "Pilot study of locomotor asymmetry in horses walking in circles with and without a rider"

_PeerJ, doi:10.7717/peerj.16373_

## Round 0.1 · original submission · Minor Revisions

Dear Authors,

As per the recommendations of our expert reviewers, the manuscript attracts few important points to be addressed. Therefore, I invite you to revise the manuscript. Please revise and resubmit asap.
All the best

·

Basic reporting

a. language was good but few sentences were poorly written, need more attention
b.References with in the text and at the end both have several mistakes so it is advisable to go through authors guidelines properly. i also mentioned details it in pdf comment boxes
c. article theme was very good
d. tables are too much and data repeated in text

Experimental design

a. sample may be more why only 15 horses and author use different breeds so data may not be authantic
b. please explain what probability of error in statistics and how overcome consider health status, age, sex environment effect etc.

Validity of the findings

please explain what probability of error in statistics as you use 15 animals of different breeds sex and how overcome consider health status, age, sex environment effect etc.

Additional comments

1. Discussion is very elaborative please concise it point to point
2. go through text comment in pdf file

Reviewer 2 ·

Basic reporting

1. The language used is of high standard and yet easy to comprehend, clear and free of ambiguity. personally i liked authors' approach to make the terms and definitions easy to understand throughout the introductory part that was useful to construct the ideological plane towards the paper.
2. Exhaustive literature survey gave enough field background
3. Figures and tables are self-explanatory
4. The pivotal crux of 'does laterality exits in horses' have been demonstrated in a fine manner

Experimental design

1. The article appears to meat the originality criterion and is within the wide scope of the esteemed journal.
2. It certainly fills the identified knowledge gap and the research question is well defined
3. I appreciate authors for their adherence with high technical and ethical standards
4. The methodology is easy to adopt and replicate, in fact, as suggested by authors sample size can be widen to test it in future for physically sound and challenged horses.
My few suggestions/ queries are as follows:
A. Whether laterality is an acquired behaviour or innate? can it differ during life time of an individual. we need to compare the laterality in horses according to age to check such concept.
B. As exemplified by feeding behaviour, can any external intervention in micro-environment of animal make the animal to choose left over right? and how easy is such transit if at all it happens?
C. Young animals are still to have epiphyses ossified, so can this affect laterality and to what extent?
In nutshell, I can see many possibilities to further this work.

Validity of the findings

1. The work certainly has novelty element and as state earlier more work needs to be done in future for the benefit of science.
2. The data sound robust and statistically sound
Well concluded and research question leads supporting results and creates more curiosity

Additional comments

videographic data would certainly help, if possible and if within the scope of metadata of journal at large

·

Basic reporting

Good

Experimental design

Good

Validity of the findings

Good

Additional comments

Authors should check for grammatical errors and English inconsistencies.
Arrange all the references as per the format of the journal.

---

## Round 0.2 · accepted · Accept

Dear authors,

As per the recommendations of our expert reviewers, the manuscript is accepted in its current format.

This is an academic acceptance and still needs a few editorial tasks to be done. Therefore, I request you to be available for the next few days to avoid any delays.

All the best for your future submissions.

·

Basic reporting

Good work, well explain

Experimental design

Optimum

Validity of the findings

Satisfied

Additional comments

Applied work

·

Basic reporting

Good.

Experimental design

Good.

Validity of the findings

Good.

Additional comments

Good.